# Short and Middle Functional Outcome in the Static vs. Dynamic Fixation of Syndesmotic Injuries in Ankle Fractures: A Retrospective Case Series Study

**DOI:** 10.3390/jcm12113637

**Published:** 2023-05-24

**Authors:** Vito Pavone, Giacomo Papotto, Andrea Vescio, Gianfranco Longo, Salvatore D’Amato, Marco Ganci, Emanuele Marchese, Gianluca Testa

**Affiliations:** 1Department of General Surgery and Medical-Surgical Specialties, Section of Orthopaedics and Traumatology, A.O.U. Policlinico Rodolico—San Marco, University of Catania, 95123 Catania, Italy; vitopavone@hotmail.com (V.P.);; 2Department of Orthopedic Surgery, Trauma Center, Cannizzaro Hospital, 95100 Catania, Italy

**Keywords:** ankle fractures, syndesmotic injury, dynamic fixation, trans-syndesmotic screw

## Abstract

Background: Syndesmotic injuries are common lesions associated with ankle fractures. Static and dynamic fixation are frequently used in syndesmotic injury-associated ankle fractures. The purpose of this study is to compare short- and mid-term quality of life, clinical outcomes, and gait after static stabilization with a trans-syndesmotic screw or dynamic stabilization with a suture button device. Methods: Here, 230 patients were enrolled in a retrospective observational study. They were divided in two groups according to the fixation procedure (Arthrex TightRope^®^, Munich, Germany) synthesis vs. osteosynthesis with a 3.5 mm trans-syndesmotic tricortical screw). They then underwent clinical assessment using the American Foot and Ankle Score (AOFAS) at 1, 2, 6, 12, and 24 months after surgery. Quality of life was assessed according to the EuroQol-5 Dimension (EQ-5D) at 2 and 24 months after surgery in the follow-up; gait analysis was performed 2 and 24 months postoperatively. Results: Significant differences were found at a two-month follow-up according to the AOFAS (*p* = 0.0001) and EQ-5D (*p* = 0.0208) scores. No differences were noted in the other follow-ups (*p* > 0.05) or gait analysis. Conclusion: The dynamic and static fixation of syndesmotic injuries in ankle fracture are both efficacious and valid procedures for avoiding ankle instability. The suture button device was comparable to the screw fixation according to functional outcomes and gait analysis.

## 1. Introduction

Ankle fractures are one of the most common injuries and often require surgical treatment to restore the anatomic congruity of the ankle mortise to provide stable load transmission and to ease rehabilitation [1,2,3,4]. In ankle fractures, syndesmotic injury occurs in approximately 50% of type Weber B and in all type Weber C fractures [5]. Thus, there are two common situations in which the syndesmosis is compromised according to the Lauge–Hansen classification: pronation external rotation (PER) fractures and supination external rotation fractures (SER) [5]. 

Syndesmosis injury could result in syndesmosis instability due to an SER fracture mechanism according to Stark and colleagues. Their study found that 39% of SER fractures with deltoid ligament rupture showed diastasis on stress testing [6]. Syndesmotic injuries can also be seen in 13% to 20% of ankle fracture patterns [7], Maisonneuve fracture [8,9], and posterior malleolar fractures [10]. Isolated syndesmosis injuries are common in the competitive athletic population [11,12]. The metallic trans-syndesmotic screw has been the gold standard for stabilizing unstable syndesmosis [13,14,15]. More recently, dynamic fixation such as the suture button device, especially the suture button device, has been reported with some potential advantages, including allowing physiological movement while retaining the required reduction. Flexible fixation also offers less risk of implant removal and recurrent syndesmotic diastasis as well as earlier rehabilitation [16,17,18]. 

Biomechanical investigations have demonstrated that the strength of the suture button device is comparable to that of a tricortical 3.5 mm syndesmotic screw [19,20]. However, the international literature has only a few high-quality studies that assess the quality of life and gait after syndesmosis fixation. Doll et al. [21] proposed a study protocol to assess the differences in gait analysis and clinical outcome after suture button device or screw fixation in acute syndesmosis rupture, but the trials have not yet been published. A pedobarographic analysis supports the nonsuperiority of a device when compared to the other tools [22,23]. The purpose of our study is to compare the short- and mid-term quality of life, clinical outcomes, and gait after static stabilization with a trans-syndesmotic screw or dynamic stabilization with a suture button fixation device (Arthrex TightRope^®^). It has been hypothesized that dynamic stabilization could thus reduce recovery time due to the removal surgery in trans-syndesmotic screws but not the patient’s quality of life.

## 2. Materials and Methods

### 2.1. Demographic Data

Of the 312 patients in our study, 26.4% were lost to follow up. There are no differences between these patients and the remaining cohort. Thus, 230 patients with syndesmosis injury associated with bimalleolar and Maisonneuve fractures were treated and analyzed from December 2015 to December 2019 (Figure 1). The exclusion criteria were pediatric ankle fractures, isolated syndesmosis injuries, tibial plafond fractures, Weber type A fractures, trimalleolar fractures, open fractures, and pathological ankle fractures. Our cohort was aged 40.8 ± 13.2 years (range 18–63); 140 patients (60.9%) were male and 90 patients (39.1%) were female. In 129 patients (56.5% of cases), the right side was affected, and 101 patients (43.5%) had a left-side injury. The BMI average was 23.8 ± 4.7 (Table 1).

### 2.2. Classification Systems

Danis–Weber classification divides fractures into three groups considering the peroneal fracture with respect to syndesmosis. In type A, the peroneal malleolus fracture is below the syndesmotic level and is due to SA. In type B, the fracture is at the syndesmosis and is due to SER or PA. The malleolar fracture is above the syndesmosis in type C and is due to PER [24]. According to Denis–Weber classification, 119 patients (51.7%) were type B, and 111 patients (48.3%) were type C.

### 2.3. Surgical Techniques

All of our patients were treated for lateral malleolus fixation with a 1/3 tubular plate with 3.5 mm screws as well as an interfragmentary screw. The tibial malleolus was synthesized with two 3.5 mm half-thread cannulated screws. In Weber B with small medial fragments, the synthesis was performed with two 2.5 mm half-thread cannulated screws. The syndesmosis was stabilized in all Weber type C (PER) cases. In the Weber type B (SER) fractures, the syndesmosis was synthesized after performing lateral and medial malleoli osteosynthesis and stress tests such as the Hook Test and the Cotton Test with positive results. The result was considered positive when there were 2 mm of diastasis of the syndesmosis during the stress tests assessed intraoperatively under fluoroscopy. 

The syndesmosis was synthesized in GROUP 1 with a 3.5 mm trans-syndesmotic tricortical screw through the hole of the fibula’s plate; the screw was removed after 45 days via outpatient surgical treatment. No weight-bearing was allowed until the screw removal. In GROUP 2, the syndesmosis was stabilized with a dynamic Tight-Rope synthesis without using a force calibrator. A cast immobilization was packed for 15 days post-operatively to facilitate the healing of the soft tissues. The two cohorts underwent the same post-operative protocol. The surgeries were performed by a four-surgeons team experienced in ankle trauma.

### 2.4. Outcomes Evaluation

Follow-ups were performed at 1, 2, 6, 12, and 24 months after the surgical treatment. At each follow-up, patients were studied with standard X-ray radiograms with antero-posterior, lateral, and Mortise views; they also underwent physical and clinical examinations using the American Foot and Ankle Score (AOFAS). Gait analysis was performed with the Oxford Foot Model (OFM) [25] that used multi-segment kinematics. A 12-camera VICON 612 system (Vicon Motion Systems Ltd., Oxford, UK) was used to collect the 3D kinematics of one foot as well as both lower limbs of each subject at 100 Hz. A static standing trial was performed to define the segment axes before three markers were removed for the walking trials. Subjects were asked to walk at their usual speed along a 10 m walkway. These trials were identified visually by looking at all traces from the session (average 20 trials). The following motions were determined: the hindfoot relative to the tibia (Hindfoot/Tibia), the forefoot relative to the hindfoot (Forefoot/Hindfoot), and the forefoot relative to the tibia (Forefoot/Tibia) [26]. Gait analysis was performed at 2 and 24 months after the surgery for GROUP 1 and GROUP 2.

The evaluation of the clinical and psychological conditions of the patients used the EuroQol-5 Dimension (EQ-5D) that was created as a generic measurement instrument for measuring the quality of life and its ease of use in self-administration. The EQ-5D consists of two separate sections: In the first one, there is a subjective assessment of five dimensions (mobility, self-care, daily activities, pain/anxiety, and anxiety/depression). Every item provides the possibility to choose a level of severity. Each item ranges from 1 to 3. Level 1 represents no problem, while level 3 indicates extreme limitations. The aggregation of answers forms a five-digit number, which represents the state of health of the respondent. The three levels of response for each of the five items offer a maximum of 243 possible descriptions of the health status and allow one to highlight the presence/absence of possible problems and their intensity. Finally, an algorithm can calculate a synthetic score (EQ-5D Index) of the perceived health status. The implementation of this algorithm provides that each dimension of the health status is associated with a specific weight, as calculated for the general population using techniques from cost-utility analyses [27]. The EuroQol-5 Dimension (EQ-5D) was performed at 2 and 24 months for every patient of GROUP 1 and GROUP 2. Clinical assessment data were electronically collected by the same authors (A.V. and S.D.A.).

### 2.5. Statistical Analysis

Continuous data are presented as the mean and standard deviation, as appropriate. The analysis of variance test and Tukey–Kramer method were used to compare the AOFAS, EQ-5D, and gait analysis parameters. The selected threshold for significance was *p* < 0.05. The estimated sample size for this study was obtained using the Bernoulli model with a z-score = 95%. All statistical analyses were performed using the 2016 GraphPad Software (GraphPad Inc., San Diego, CA, USA).

## 3. Results

The syndesmosis injuries were treated according to two different devices: one group was treated with a tricortical trans-syndesmotic screw (GROUP 1); the other group was treated with suture button fixation, TightRope (GROUP 2). The demographic characteristics of the sample are reported in Table 1. According to the demographics, no statistical differences were found between the cohorts (*p* > 0.05).

At a one-month follow-up, the Group 1 mean AOFAS score was 50.7 ± 6.4, while the corresponding Group 2 mean score was 51.7 ± 6.3 (*p* > 0.05). At two months, the average AOFAS score of Group 1 was 59.5 ± 11.01, while that of Group 2 was 66.7 ± 11.8 (*p* = 0.0001). At 24 months, the average AOFAS score of Group 1 was 94 ± 2.4, and that of Group 2 was 94.7 ± 1.5 (Table 2). The difference in mean EQ-5D scores was found to be significant at 2 months of follow-up; these scores were similar at 24 months of follow-up (Table 2).

At a two-month follow-up, significant differences were found for the following parameters: hindfoot/tibia dorsiflexion (*p* = 0.03), forefoot/tibia supination (*p* = 0.03), and forefoot/tibia adduction (*p* = 0.03). No other significant differences were found according to the gait analysis parameters (*p* > 0.05) at the two-month and 24-month follow-up (Table 3 and Table 4).

In total, we reported 18 complications: 13 with wound dehiscence with superficial infection and 5 with wound necrosis. In the case of superficial infection, a second operation was performed with surgical wound debridement, biopsy samples, and targeted antibiotic therapy. In the second case, outpatient debridement and the placement of vacuum-assisted continuous-therapy (VAC-therapy) was performed (Table 5). 

## 4. Discussion

Suture button device-treated patients at 2 months of follow-up had a faster recovery and higher foot dorsiflexion and forefoot supination and adduction than the trans-syndesmotic screw fixation cohort. The faster recovery in dynamic fixation was associated with a higher quality of life two months after the surgery. The procedures’ functional outcomes were comparable six months after surgery. Syndesmotic lesions are widespread clinical conditions and can occur with concomitant ankle fractures or without fractures; however, the latter is extremely rare [28,29].

The treatment of the syndesmotic complex injury is necessary to avoid chronic instability [30]. Although the current gold standard for the treatment of syndesmotic lesions is the fixation of the syndesmotic screw, the use of the suture button technique has aroused interest and has increased rapidly over the last decade [31]. There is a strong debate in the literature about the screw placement and whether and when to remove it. Screw abandonment causes it to rupture because of the physiological movements of the syndesmosis between the tibia and fibula, although a low percentage of patients have experienced syndesmotic malreductions after it breaks. There are no absolute indications in the literature on the timing of trans-syndesmotic screw removal. Early removal of the syndesmotic screw before ligament healing can lead to instability and diastasis of the syndesmosis as well as an increase in complications related to the second surgery such as infection. This, in turn, could increase the recovery time and pain while harming the psychological state of the patients [32,33,34,35]. The use of a dynamic fixation such as a suture button device allows for physiological movements, and it does not require further intervention for its removal. In our study, a better functional outcome was recorded at two months of follow-up; in fact, the average AOFAS score of Group 1 was 59.5 ± 11.01, while that of Group 2 was 66.7 ± 11.8 (*p* = 0.0001). The weight bearing ban in patients treated with a tricortical screw has been assumed as the principal reason for the data. Moreover, the second surgery need for the screw removal could partially delay the physical recovery and provide a negative effect directly on the gait and indirectly on the quality of life.

Lubberts et al. [35] performed cadaver biomechanical studies and argued that the stabilization of the syndesmosis with a suture button device seems to stabilize the coronal plane but not the movement in the sagittal plane. This outcome can be explained considering that the suture button device is placed in dynamic compressions on the coronal plane, but the construct is installed through perforated channels that far exceed the diameter of the sutures of the suture button device [35]. This facilitates a considerable residual sagittal instability because of the persistence of diameter differences. LaMothe et al. [36] assessed fibular motion in a cadaveric model after fixation with a tetra-cortical 4.0 mm screw or a single suture-button construct using fluoroscopy, as validated by a four-camera motion-capture system. They found that the screw or suture-button fixations could constrain coronal plane fibular motion in response to an external rotation stress test. In contrast, Ebramzadeh et al. [37] and Forsythe et al. [38] observed that a single suture button construct could maintain syndesmotic stability in the coronal plane. Adding to these discrepancies, Soin et al. found that two suture button constructs provided similar syndesmotic stability in the coronal and sagittal planes compared to a single tetra-cortical syndesmotic screw, but neither restored native motion [39]. Teramoto et al. [36] found that one or two suture button constructs were not able to restore stability in both the coronal and sagittal planes compared to the intact syndesmosis or fixation with a tetra-cortical syndesmotic screw. 

In our series, the ankle stability was intraoperatively clinically assessed, and malreductions or instabilities were recorded. At the same time, no further radiological or tomographic evaluations were performed; the literature includes studies of screw syndesmosis fixation using bilateral CT to evaluate the reduction in the syndesmosis. These studies have reported malreduction rates of 15–44% in 1.5–8.4 years of follow-up [40,41,42]. Most studies evaluating suture button device fixation for unstable syndesmosis reported malreduction rates of 0% but used only simple radiography to assess malreduction [17,43,44,45,46,47,48]. Only the study by Treon et al. [49] reported a syndesmotic malreduction rate of 11% when a suture button device was used. Naqvi et al. [47] compared the trans-syndesmotic screw and suture button device fixation with CT scans of both ankles to assess the reduction in the syndesmosis and found no malreduction with suture button device fixation in 23 patients after a follow-up of at least 18 months. Anand et al. published a multicenter case series consisting of 36 patients. They demonstrated that the ankle suture button device maintained satisfactory reduction in the ankle mortise in 97% of cases, with a mean follow-up of 14 months [50]. Sagi and colleagues [42] used CT and clinical follow-up at a minimum of two years from fracture fixation. They showed that this strategy produces significant improvements in terms of reducing syndesmosis. In conflict with this evidence, a recent study reported questionable advantages in assessing the quality of distal tibial-fibula joint reduction when a suture-button system is used due to a considerable rate of false-negatives [51].

The dynamic nature of the push-button suture device could theoretically allow for a certain degree of physiological micro-mobility of the syndesmosis, leading to an earlier return, full weight bearing, and a better objective range of motion measurements. Screw fixation does not allow for normal movement of the syndesmosis during healing; if stabilization is not achieved or if the load is early, the screw breaks or the implant is mobilized. Thornes et al. [47] noted that patients in the suture button group were maintained without weight bearing for a significantly shorter mean time than patients in the syndesmotic screw group (4.1 weeks versus 6.3 weeks, *p* = 0.01), with no patient in the suture button group requiring implant removal. Degroot et al. [45] reported an average lift time at full load of 5.7 weeks with TightRope, with no signs of implant failure or residual displacement in a follow-up of 20 months. Cottom et al. and Thornes et al. demonstrated that full and fast loading could lead to accelerated rehabilitation [52,53]. Interestingly, some included studies reported that patients in the dynamic fixation group appeared to have less pain and discomfort, which may also contribute to faster, full weight bearing [54,55,56]. It could be assumed that a faster recovery could improve the quality of life in ankle fracture patients treated with dynamic fixation. The functional limitation due to physical therapy interruption and the additional surgery in trans-syndesmotic screw patients could cause a negative interpretation of their clinical condition and backsliding during physical rehabilitation. To the best knowledge of the authors, no previous study aimed to assess the comparison between the two procedures according to the gait analysis performed. In 2020, Doll et al. [21] proposed a study protocol for a prospective randomized pilot study with the purpose of comparing the monitor ankle range of motion and maximum ankle power in gait as functional outcome parameters of instrumented gait analysis, as well as the clinical and radiographic outcome for assessing the stabilization of acute syndesmosis rupture. We strongly encourage the participation of these kinds of trials and study groups. The study accounts for different limits: the small sample size is the principal. Nodal irritation or secondary extension and radiologically visible osteolysis are the principal complications of suture button devices; due to the reduced number of participants, the study was not able to demonstrate this problem. Despite ankle fractures being common injures and the fact that several trials are available, in our study, the midterm follow-up and the recurrent functional assessment in addition to gait analysis reduced the higher recruitment possibility. At the same time, the retrospective nature of the review could limit the proper evaluation. Prospective randomized clinical trials are strongly encouraged. The strengths of the study are the quality-of-life evaluation and gait analysis; both typologies of syndesmosis fixation are not common for this kind of lesion, and rarely were the measurements reported in the same trial. Moreover, our findings could help the surgeons and patients make an appropriate choice.

## 5. Conclusions

The dynamic and static fixation of syndesmotic injuries in ankle fracture are both efficacious and valid procedures for avoiding ankle instability. The suture button device was comparable to the screw fixation according to functional outcomes and gait analysis.

## Figures and Tables

**Figure 1 jcm-12-03637-f001:**
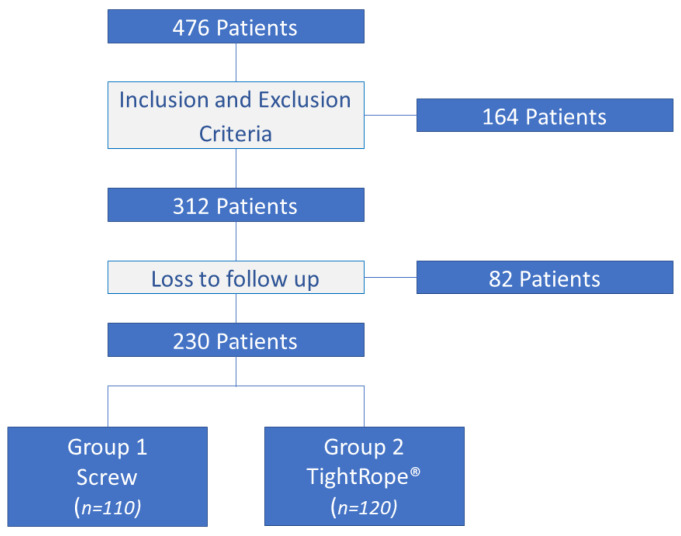
Study flowchart.

**Table 1 jcm-12-03637-t001:** Patient characteristics and baseline.

Characteristics	Group 1—Trans-Syndesmotic Screw (*n* = 110)	Group 2—TightRope (*n* = 120)	*p*-Value
Age (years)	41.2 ± 14.56	35.68 ± 11.57	0.34433
Gender (Male/Female)	64/46	76/44	0.66
Side (Left/Right)	44/66	57/63	0.08537
Weber (B/C)	57/53	62/58	0.74
Average BMI	24.2 ± 4.6	23.9 ± 4.6	0.72

**Table 2 jcm-12-03637-t002:** Group results according the AOFAS and EQ-5D scores.

	Trans-Syndesmotic Screw Group 1	Suture Button Device Group 2	*p*-Value
AOFAS 4 Weeks	50.7 ± 6.42	51.71 ± 6.30	0.9997
AOFAS 2 Months	59.5 ± 11.01	66.73 ± 11.76	0.0001
AOFAS 6 Months	84.6 ± 8.93	87.92 ± 8.38	0.4936
AOFAS 12 Months	89.1 ± 7.01	93.08 ± 5.06	0.2318
AOFAS 24 Months	94.0 ± 2.39	94.69 ± 1.46	1.00
EQ-5D 2 Months	0.0 ± 0.44	0.18 ± 0.41	0.0208
EQ-5D 24 Months	0.8 ± 0.06	0.87 ± 0.04	0.6701

**Table 3 jcm-12-03637-t003:** Gait analysis results at two months.

	Trans-Syndesmotic Screw Group 1	Suture Button Device Group 2	*p*-Value
hindfoot/tibia inversion	3.2 ± 2.8	4.1 ± 1.6	0.0562
hindfoot/tibia dorsiflexion	6.2 ± 2.4	7.4 ± 2.9	0.0296
hindfoot/tibia rotation	10.6 ± 2.2	11.3 ± 2.9	0.1860
forefoot/hindfoot supination	10.4 ± 3.3	11.5 ± 2.4	0.0649
forefoot/hindfoot dorsiflexion	23.3 ± 4.1	24.2 ± 4.6	0.3142
forefoot/hindfoot adduction	7.8 ± 3.1	9.4 ± 4.8	0.0721
forefoot/tibia supination	7.2 ± 3.3	8.5 ± 2.4	0.0297
forefoot/tibia dorsiflexion	11.6 ± 3.1	11.8 ± 3.3	0.7603
forefoot/tibia adduction	8.8 ± 3.1	10.2 ± 3.2	0.0320

**Table 4 jcm-12-03637-t004:** Gait analysis results at 24 months.

	Trans-Syndesmotic Screw Group 1	Suture Button Device Group 2	*p*-Value
hindfoot/tibia inversion	4.9 ± 1.8	4.7 ± 1.6	1.00
hindfoot/tibia dorsiflexion	7.7 ± 3.0	8.2 ± 2.9	0.9998
hindfoot/tibia rotation	11.1 ± 4.2	12.0 ± 2.9	0.9847
forefoot/hindfoot supination	10.7 ± 4.3	12.7 ± 3.4	0.3039
forefoot/hindfoot dorsiflexion	24.3 ± 7.1	26.3 ± 5.6	0.3039
forefoot/hindfoot adduction	10.2 ± 4.7	10.7 ± 3.8	0.9965
forefoot/tibia supination	7.8 ± 2.3	9.1 ± 3.4	0.5786
forefoot/tibia dorsiflexion	13.6 ± 2.1	12.8 ± 2.3	0.9462
forefoot/tibia adduction	10.8 ± 3.6	11.2 ± 4.2	0.9992

**Table 5 jcm-12-03637-t005:** Postoperative complications.

	Trans-Syndesmotic Screw Group 1	Suture Button Device Group 2
wound necrosis	3	2
superficial infections	5	8

## Data Availability

The data presented in this study are available in text and in tables.

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
