# Peer review of "Short and Middle Functional Outcome in the Static vs. Dynamic Fixation of Syndesmotic Injuries in Ankle Fractures: A Retrospective Case Series Study"

_jcm, 2023, doi:10.3390/jcm12113637_

Round 1
Reviewer 1 Report (New Reviewer)
- improve title (short- and mid-term...)
- improve grammar in several parts of the manuscript
Introduction:
- specify "lateral component", what do you mean by "syndesmosis", do you include lateral ligaments? Be more concise
- syndesmotic injury is not the same as syndesmotic instability and has to be differentiated properly
- dynamic stabilization, not flexible
- you mention potential advantages of the suture button, I think there is enough evidence for efficiacy, safety and improved outcome and we are beyond the point of potential, there are advantages!
- add Reference in line 60
- clearly state your hypothesis
M&M
- again: syndesmotic injury? what does that mean? It does not mean that there is instability
- explain why trimall fx were excluded
- I am not aware of the terminology "peroneal malleolus fracture" or "tibial mallelous", I know what you mean but it is not common terminology and has to be improved
- why stabilize all Weber C fx? It is known (literature research!) that not every Weber C needs syndesmotic stabilization, did you test those as well intraop?
- 2mm diastasis? how much is acceptable? 1mm? 2 is too much? Reference? What is the physiological motion? How did you measure 2mm?
- did you ever had to do a medial incision for the medial button? It can happen that soft tissue is interposed, any problems with that? be more concise
- same postop protocol (lines 115/116)? so NWB for 6 weeks for both groups?
- how did you assess reduction intraop? did you use a clamp? at which height were screws and SB placed and why?
- xrays postop weightbearing?
Results:
- lesion restored? improve phrasing
- EQ-5D better after 2 months, how would you explain that and is it relevant if both groups were NWB for 6 weeks?
- gait analysis is nice to have, what kind of limitations would you expect though? You should phrase expected or possible gait limitations and alterations and put your results in context to those
Discussion:
- ad references (lines 207-211)
- did you not remove any SB device?
- line 239/240: how were malreductions recorder? Did you have any??
- lack of literature, study by Sanders (RCT) has to be cited when you talk about SB outcomes and look at the study by Spindler and Baumbach looking at correction potential of SB
Author Response
- improve title (short- and mid-term...)
done
- improve grammar in several parts of the manuscript
done
Introduction:
- specify "lateral component", what do you mean by "syndesmosis", do you include lateral ligaments? Be more concise
The paragraph was removed
- syndesmotic injury is not the same as syndesmotic instability and has to be differentiated properly
The phrase was re-written. According to Barbachan Mansur et al.(https://doi.org/10.1016/j.fcl.2023.01.006) “distinguishing between injury and instability remains challenging..” and not clear definition is present in literature. In the manuscript the syndesmotic injury is the lesion related to the ankle fracture and may result in sindesmotic instability
- dynamic stabilization, not flexible
The term was modified
- clearly state your hypothesis
The hypothesis was partially re-written
M&M
- again: syndesmotic injury? what does that mean? It does not mean that there is instability
See previous answer
- explain why trimall fx were excluded
The inclusion of the trimalleolar fractures could generate an assessment bias related to the posterior malleolus fracture. For this reason the lesion were exluded
- I am not aware of the terminology "peroneal malleolus fracture" or "tibial mallelous", I know what you mean but it is not common terminology and has to be improved.
The terms were changed
- why stabilize all Weber C fx? It is known (literature research!) that not every Weber C needs syndesmotic stabilization, did you test those as well intraop?
Weber C fracture are the result of syndesmosis injury and the interosseous membrane lesion. For this reason the authors perform the syndesmosis stabilization in all cases.
- 2mm diastasis? how much is acceptable? 1mm? 2 is too much? Reference? What is the physiological motion? How did you measure 2mm?
As described the 2mm more than phisiological diastasis was define as cut-oof in our study.
- did you ever had to do a medial incision for the medial button? It can happen that soft tissue is interposed, any problems with that? be more concise
Tight-Rope surgical procedure include a medial incision for the button. The correct skeletonization prevent the soft tissue interposition
- same postop protocol (lines 115/116)? so NWB for 6 weeks for both groups?
As report
- how did you assess reduction intraop? did you use a clamp? at which height were screws and SB placed and why?
According to the fluroscopic image in LL, PA and mortise and dynamic twst
- xrays postop weightbearing?
No
Results:
- lesion restored? improve phrasing
Done
- EQ-5D better after 2 months, how would you explain that and is it relevant if both groups were NWB for 6 weeks?
Eq-5D is quality of life tool assessment and it is patient related score. As reported in discussion, the need of second surgery could influence the patients quality of life
- gait analysis is nice to have, what kind of limitations would you expect though? You should phrase expected or possible gait limitations and alterations and put your results in context to those
No prevoius studies on gait analysis were published, in the authors opinion the gait analysis had no more limit than the rpesent in proper section
Discussion:
- ad references (lines 207-211)
Added
- did you not remove any SB device?
In our series no Sutur button were removed despite the 10 complicantions
- line 239/240: how were malreductions recorder? Did you have any??
No post-operative mal reduction were recorded
- lack of literature, study by Sanders (RCT) has to be cited when you talk about SB outcomes and look at the study by Spindler and Baumbach looking at correction potential of SB
The studies were discussed and the Spinder qarticle included in discussion
Reviewer 2 Report (New Reviewer)
Journal of Clinical Medicine
Title: Short and middle functional outcome in static vs dynamic fixation of syndesmotic injuries in ankle fractures: a retrospective case series study.
Vito Pavone1, Giacomo Papotto2, Andrea Vescio1, Gianfranco Longo2, Salvatore D’Amato1, Marco Ganci2, 5 Emanuele Marchese1, Gianluca Testa1*
Abstract: Background: Syndesmotic injuries are common lesions associated with ankle fractures. Static and dynamic fixation are frequently used in syndesmotic injury-associated ankle fractures. The purpose of this study is to compare short- and mid-term quality of life, clinical outcomes, and gait after static stabilization with a trans-syndesmotic screw or dynamic stabilization with a suture button device. Methods: Here, 230 patients were enrolled in a retrospective observational study. They were divided in two groups according to the fixation procedure (Arthrex TightRope® synthesis vs osteo-synthesis with a 3.5-mm trans-syndesmotic tricortical screw). They then underwent clinical assessment using the American Foot and Ankle Score (AOFAS) at 1, 2, 6, 12, and 24 months after surgery. Quality of life was assessed according the EuroQol-5 Dimension (EQ-5D) at 2 and 24 months after surgery in follow-up; gait analysis was performed 2 and 24 months postoperatively. Results: Significant differences were found at two- month follow-up according to the AOFAS (p=0.0001) and EQ-25 5D (p=0.0208) scores. No differences were noted in the other follow-ups (p>0.05) or gait analysis. Conclusion: Dynamic and static fixation of syndesmotic injuries in ankle fracture are both efficacious and valid procedures for avoiding ankle instability. The suture button device was comparable to the screw fixation according to functional outcomes and gait analysis.
Abbreviations: Appropriate use of few abbreviations.
Purpose:
The purpose of this study is to compare short- and mid-term quality of life, clinical outcomes, and gait after static stabilization with a trans-syndesmotic screw or dynamic stabilization with a suture button device.
Measurement tools:
AOFAS, 152 EQ-5D, and gait analysis parameters.
The reviewer appreciates the authors’ explanations about these measurement scales. In the future I suggest you report whether data were normally distributed. Were there ceiling or floor effects?
Statistics:
The analysis of variance test and Tukey–Kramer method were used to compare the AOFAS, EQ-5D, and gait analysis parameters. Non-parametric tests might produce differences you aren’t seeing with the parametric ANOVA. Do you think the data meet the criteria for parametric analysis?
Table 2. At two months, the average AOFAS 164 score of Group 1 was 59.5 ± 11.01 while Group 2 was 66.7 ± 11.8 (p= 0.0001). Also EQ-5D at 2 months difference was significant at 0.02
Table 3. At two-month follow-up, significant differences were found for the following parameters: hindfoot/tibia dorsiflexion (p=0.03), forefoot/tibia supination (p=0.03), and forefoot/tibia adduction (p=0.03).
Discussion:
“Suture button device-treated patients at 2 months follow-up had faster recovery and higher foot dorsiflexion and forefoot supination and adduction than the trans-syndesmotic screw fixation cohort. Faster recovery in dynamic fixation was associated with a higher quality of life two months after the surgery. The procedures’ functional outcomes were comparable six months after surgery.” ….
The Conclusion differs from what is written in the Abstract. I believe the Abstract Conclusion is more justified.
Otherwise, add a qualifying term (2 months)
5. Conclusions:
“Dynamic and static fixation of syndesmotic injuries in ankle fracture are effective and valid procedures to avoid ankle instability. In this series, at short-term follow-up”
Reviewer suggests adding (2 months) here,
“suture button device reported better results than the static screw fixation, according to the functional outcome, quality of life and gait analysis.”

Author Response
Thank you for your evaluation,
conclusion are changed. Your suggestion has been applied to manuscript.
Reviewer 3 Report (New Reviewer)
1. The introduction is a bit long, I think that removing the first paragraph and retaining the other 3 paragraphs would be sufficient.
2. Are you able to provide an account on the characterisics of the the patients who were lost to follow-up? It could be beneficial to add a sentence that there are (or there are none) differences between these patients and the remaining cohort.
3. In table 1 please add BMI in group1 and BMI in group 2
4. It is absolutely necessary to show how many of the fractures in each group were Weber B and how many of them were Weber C.
5. Although Lauge-Hanssen classification is mentioned and referenced in both the introduction and methods, the distribution of types of fractures between the groups is not provided. If you have this information please provide it as well. Otherwise mentioning the classification in methods and the introduction would be inappropriate.
6. It is important to specify the indications for screw vs rope fixation. Was it fracture type or surgeon's preference?
7. Table 5 - please add column names "Transsyndesmotic screw" etc
8. The conclusions in the text differ from the conclusions in the abstract. Anyway, I will be able to say anything regarding the validity of the conclusions after receiving the above corrections.
Thank you
Author Response
Q1) The introduction is a bit long, I think that removing the first paragraph and retaining the other 3 paragraphs would be sufficient.
A1) The first paragraph has been removed
Q2. Are you able to provide an account on the characterisics of the the patients who were lost to follow-up? It could be beneficial to add a sentence that there are (or there are none) differences between these patients and the remaining cohort.
A2) The sentence has been added
Q3) In table 1 please add BMI in group1 and BMI in group 2
A3) Bmi has been added.
Q4. It is absolutely necessary to show how many of the fractures in each group were Weber B and how many of them were Weber C.
A4) Done
Q5) Although Lauge-Hanssen classification is mentioned and referenced in both the introduction and methods, the distribution of types of fractures between the groups is not provided. If you have this information please provide it as well. Otherwise mentioning the classification in methods and the introduction would be inappropriate.
A5) The Lauge-Hanssen classification has been eliminated.
Q6) It is important to specify the indications for screw vs rope fixation. Was it fracture type or surgeon's preference?
A6) The indication has been described
Q7) Table 5 - please add column names "Transsyndesmotic screw" etc
A7) Table 5 has been updated
Q8) The conclusions in the text differ from the conclusions in the abstract. Anyway, I will be able to say anything regarding the validity of the conclusions after receiving the above corrections.
A8) The conclusion has been changed.
Round 2
Reviewer 1 Report (New Reviewer)
N/A
This manuscript is a resubmission of an earlier submission. The following is a list of the peer review reports and author responses from that submission.
Round 1
Reviewer 1 Report
This study focuses on the stabilization technique for ankle fractures with syndesmosis injury. The aim of the study was to compare two stabilization techniques in terms of quality of life and clinical outcome. For this purpose, 2 questionnaires and a gait analysis were performed.
The reviewer thanks the authors for the presented study. In this context, please allow me to make the following comments:
The treatment of syndesmosis injuries by means of TightRope or set screw has been scientifically reviewed very well in the last 10 years and especially prospective randomized studies from the last 5 years could not show any significant difference between both procedures in the medium term (1-2 year follow up) as well as in the long term (10 year follow up). Therefore, it is difficult for me to find a new scientific approach in the present study.
With regard to the references, the main level I studies also published in the JCM are missing.
From a scientific point of view, I was pleased about the performance of a gait analysis and thank the authors for this new approach. I would have liked to see this not only after 2 and 24 months in the study, but in further assessments over the 2 years. Based on only 2 measurement points, the authors made a scientific statement that I can't share like that. The first measurement point was performed directly after the removal of the screw and can therefore only be used as a reference to a limited extent. The 2nd measuring point was chosen only 2 years later, so that no statement can be made as to how the two groups behaved within the 2 years.
Therefore, from my point of view, such a conclusion cannot be drawn after only one really valid measuring point after 24 months. Much more informative would have been a close-meshed gait analysis and evaluation of the EuroQol-5 dimension to show a real difference in quality of life and rehabilitation time over the time axis. Unfortunately, the authors do not provide us with this scientific information. Instead, the reader receives information that has already been presented several times in Level I studies over the last 6 years.
Comments in detail:
Abstract:
Line 19: What kind of Tight Rope was used? (knotless or TightRope with knot)
Line 22: why only after 2 and 24 months ?
Instruction:
Be sure to shorten. Anatomical basics should be clear to the reader and do not need to be explained at this level.
Important Level I studies on this topic from the last 7 years are missing. The reviewer recommends the authors to review the literature again.
Line 69/70: This is not a new approach and has already been sufficiently explained scientifically. The authors should rather focus on gait analysis, because that is the actual new approach.
Material and Methods:
What about isolated syndesmosis injuries and Maisonneuve fractures ?
Classifications are known to the readers. Please shorten.
Was there an ethics vote ?
How high was the LFU (Lost to follow up)
How was the screw removed ? Outpatient ? Inpatient ?
Why was the screw removed after only 45 days?
How was the postoperative protocol ? Only 2 weeks partial loading and then full weight bearing?
What about fracture healing , what about the screw?
Were there any screw breakages ?
How many surgeons?
Line 192
Please name groups in the tables
Line 202: 8% complication rate? Seems a bit high if only closed fractures were treated.
How many knot irritations in the TightRope Group?
Line 205-209
When did the infections occur? When were the revision operations performed?
Discussion:
Line 219-227 : should be part of the instruction section
Line 228: Correct statement , but then why was the screw removed after 6 weeks in the study ?
Line 238-253 Please do not only quote but also dicuss in front of your own results
Line 256-268: Please again do not only quote but also dicuss in front of your own results, malreduction was not subject of the study and was also only controlled by x ray and not by CT postoperatively. Please explain why. Please also explain how you checked and excluded a malreduction intraoperatively.
286 That is correct: why did the authors then only perform a gait analysis shortly after removal of the screw? This also explains why the TightRope group was better at this point. Please discuss this point in detail.
Line 287: These fractures were excluded according to the study protocol. why is the care of these fractures listed here ?
Conclusion:
The conclusion based on these data is not only daring but scientifically very questionable because there were only 2 real measurement points after 2 and 24 months, whereby the first measurement point is only conditionally usable due to the 2nd operation on the screw. To draw conclusions on this basis for the entire 24 months is purely speculative.
References:
incomplete
Author Response
This study focuses on the stabilization technique for ankle fractures with syndesmosis injury. The aim of the study was to compare two stabilization techniques in terms of quality of life and clinical outcome. For this purpose, 2 questionnaires and a gait analysis were performed.
The reviewer thanks the authors for the presented study. In this context, please allow me to make the following comments:
The treatment of syndesmosis injuries by means of TightRope or set screw has been scientifically reviewed very well in the last 10 years and especially prospective randomized studies from the last 5 years could not show any significant difference between both procedures in the medium term (1-2 year follow up) as well as in the long term (10 year follow up). Therefore, it is difficult for me to find a new scientific approach in the present study.
With regard to the references, the main level I studies also published in the JCM are missing.
From a scientific point of view, I was pleased about the performance of a gait analysis and thank the authors for this new approach. I would have liked to see this not only after 2 and 24 months in the study, but in further assessments over the 2 years. Based on only 2 measurement points, the authors made a scientific statement that I can't share like that. The first measurement point was performed directly after the removal of the screw and can therefore only be used as a reference to a limited extent. The 2nd measuring point was chosen only 2 years later, so that no statement can be made as to how the two groups behaved within the 2 years.
Therefore, from my point of view, such a conclusion cannot be drawn after only one really valid measuring point after 24 months. Much more informative would have been a close-meshed gait analysis and evaluation of the EuroQol-5 dimension to show a real difference in quality of life and rehabilitation time over the time axis. Unfortunately, the authors do not provide us with this scientific information. Instead, the reader receives information that has already been presented several times in Level I studies over the last 6 years.
Comments in detail:
Abstract:
Instruction:
Q3) Be sure to shorten. Anatomical basics should be clear to the reader and do not need to be explained at this level.
A3) thank for your comment, the paragraph was resume
Q4) Important Level I studies on this topic from the last 7 years are missing. The reviewer recommends the authors to review the literature again.
Q5) Line 69/70: This is not a new approach and has already been sufficiently explained scientifically. The authors should rather focus on gait analysis, because that is the actual new approach.
A4 e A5) new references were added according to reviewers comments.
Material and Methods:
Q6) What about isolated syndesmosis injuries and Maisonneuve fractures ?
A6) Isoled syndesmosis injures were not included
Q7) Classifications are known to the readers. Please shorten.
A7) done
Q8) How high was the LFU (Lost to follow up)
A8) lost to follow up was included
Q9) How was the screw removed ? Outpatient ? Inpatient ?
A9) the screw removal was performed in outpatient
Q10) Why was the screw removed after only 45 days?
A10) Thank for your comment. According to our protocol the screw is removed 45 days after the implant in order to provide the syndesmotic complex recovery
Q11) How was the postoperative protocol ? Only 2 weeks partial loading and then full weight bearing?
What about fracture healing , what about the screw? Were there any screw breakages ?
A11) In the tran-syndesmotic group the weight bearing was avoid until the screw removal, while after the suture remove the partial weight bearing was allowed in the tightrope patients.
Q12) How many surgeons?
A12) the treatment was performed by the same surgeons team composed by 3 foot and ankle traumatology trained orthopedics
Q12) Line 192
Please name groups in the tables
Done
Q13) Line 202: 8% complication rate? Seems a bit high if only closed fractures were treated.
How many knot irritations in the TightRope Group?
A13) No knot irritations were recorded
Q14) Line 205-209
When did the infections occur? When were the revision operations performed?
A14) The superficial infection occurred before the suture removal and were resolved with local and systemic antibiotic therapy. The revision was performed in conservative treatment failure cases
Discussion:
Q15) Line 228: Correct statement , but then why was the screw removed after 6 weeks in the study ?
A15) Thank for your comment. According to our protocol the screw is removed 45 days after the implant in order to provide the syndesmotic complex recovery
Q16) Line 238-253 Please do not only quote but also dicuss in front of your own results
A16) The authors consider the reported data useful for the reader in order to analyze the comparison between the fixation system. At the same time, the study design can not provide evidence to support the included literature design
Q17) Line 256-268: Please again do not only quote but also dicuss in front of your own results, malreduction was not subject of the study and was also only controlled by x ray and not by CT postoperatively. Please explain why. Please also explain how you checked and excluded a malreduction intraoperatively.
A17) the hook and cotton test were intra-operatively performed, but not objective assessment were recorded
Q18) 286 That is correct: why did the authors then only perform a gait analysis shortly after removal of the screw? This also explains why the TightRope group was better at this point. Please discuss this point in detail.
A18) the two months follow up was considered proper short term evaluation period
Q19) Line 287: These fractures were excluded according to the study protocol. why is the care of these fractures listed here ?
The paragraph was removed
Conclusion:
Q20) The conclusion based on these data is not only daring but scientifically very questionable because there were only 2 real measurement points after 2 and 24 months, whereby the first measurement point is only conditionally usable due to the 2nd operation on the screw. To draw conclusions on this basis for the entire 24 months is purely speculative.
A20) the conclusion was generalized and re-written
References:
Incomplete. Added new references
Reviewer 2 Report
The topic is one of importance given the high number of presentations to health services that are related to concerns on
the prevalence and related factors of the ankle syndesmosis injuries in the population. Also, this is an interesting aim with this clinical trial that compare the static vs dynamic fixation of syndesmotic injuries in ankle fractures. I think it would be a more clear study if the following parts were revised and supplemented. These will be discussed below relative to the information of the manuscript.
General Comments:
Overall the manuscript is generally well written and is a topic of interest. However after reading it a number of times I am still left without key take-home points. I believe these points are in the results they just need to be developed.
Specific comments:
Abstract:
1) The authors state they were assess compare short and mid-term quality of life, clinical outcome and gait after static stabilization with a trans-syndesmotic screw or dynamic stabilization with a suture button fixation device (TightRope). This seems like too much of an over simplification of what was done. I do feel that it would be beneficial to explain what specifically you are looking at in relation to ankle syndesmosis injury (this also applies to the main body of the paper). Is it the development of ankle syndesmosis injury literature. This needs to be made clearer throughout the paper. (Major Compulsory Revision)
Introduction
2) The first paragraph should have a sentence or two added that better outlines why this study is important related with lateral ankle sprain patients comparison of the Plantar Fascia and Tibialis Anterior in People With and Without Lateral Ankle Sprain https://pubmed.ncbi.nlm.nih.gov/32709515/, Ultrasonography Comparison of Peroneus Muscle Cross-sectional Area in Subjects With or Without Lateral Ankle Sprains https://pubmed.ncbi.nlm.nih.gov/27793349/ and electromiography comparison of distal and proximal lower limb muscle activity patterns during external perturbation in subjects with and without functional ankle instability https://pubmed.ncbi.nlm.nih.gov/28843163/
3) In the last paragraph, the significance of the proposed word should be included highlighting why your work is important. what is the scientific contribution of this paper? it is not clear how this paper can make a significant contribution to the state of the art. (Major Compulsory Revision).
In addition, author´s hypotheses should be included in this section.
4) This methods section is poor, needs to present a better rationale for the study and the methodology employed. Also, neither appear information related with inclusion and exclusion criteria, dates, protocol. The study design is a experimental research of ramdom sampling method, where the study was conducted in the hospital or in the university center? This research adhere to reporting CONSORT guidelines? (Major Compulsory Revision).
5) Where the experiments carried out? In a hospital? In an educational institution? Provide this information.
6) Add figure 1 as a study flow chart for the readers. (Major Compulsory Revision).
7) The Discussion section is a rehashing of the results. It does not appear that the authors include much interpretation of what the study findings mean for clinical practice or research. (Major Compulsory Revision)
FInally, the conclusión is weak and too long. (Major Compulsory Revision)
Author Response
The topic is one of importance given the high number of presentations to health services that are related to concerns on
the prevalence and related factors of the ankle syndesmosis injuries in the population. Also, this is an interesting aim with this clinical trial that compare the static vs dynamic fixation of syndesmotic injuries in ankle fractures. I think it would be a more clear study if the following parts were revised and supplemented. These will be discussed below relative to the information of the manuscript.
General Comments:
Overall the manuscript is generally well written and is a topic of interest. However after reading it a number of times I am still left without key take-home points. I believe these points are in the results they just need to be developed.
Specific comments:
Abstract:
Q1) The authors state they were assess compare short and mid-term quality of life, clinical outcome and gait after static stabilization with a trans-syndesmotic screw or dynamic stabilization with a suture button fixation device (TightRope). This seems like too much of an over simplification of what was done. I do feel that it would be beneficial to explain what specifically you are looking at in relation to ankle syndesmosis injury (this also applies to the main body of the paper). Is it the development of ankle syndesmosis injury literature. This needs to be made clearer throughout the paper. (Major Compulsory Revision)
A1) Major revision has been performed
Introduction
Q2) The first paragraph should have a sentence or two added that better outlines why this study is important related with lateral ankle sprain patients comparison of the Plantar Fascia and Tibialis Anterior in People With and Without Lateral Ankle Sprain https://pubmed.ncbi.nlm.nih.gov/32709515/, Ultrasonography Comparison of Peroneus Muscle Cross-sectional Area in Subjects With or Without Lateral Ankle Sprains https://pubmed.ncbi.nlm.nih.gov/27793349/ and electromiography comparison of distal and proximal lower limb muscle activity patterns during external perturbation in subjects with and without functional ankle instability https://pubmed.ncbi.nlm.nih.gov/28843163/
A2) Done
Q3) In the last paragraph, the significance of the proposed word should be included highlighting why your work is important. what is the scientific contribution of this paper? it is not clear how this paper can make a significant contribution to the state of the art. (Major Compulsory Revision).
In addition, author´s hypotheses should be included in this section.
A3) Thanks for your comment. A section was include in the introduction as well as the hypothesis.
Q4) This methods section is poor, needs to present a better rationale for the study and the methodology employed. Also, neither appear information related with inclusion and exclusion criteria, dates, protocol. The study design is a experimental research of ramdom sampling method, where the study was conducted in the hospital or in the university center? This research adhere to reporting CONSORT guidelines? (Major Compulsory Revision).
A4) All the data are added.
Q5) Where the experiments carried out? In a hospital? In an educational institution? Provide this information.
A5) The experimental trial was carried out in the univerisity hospital of catania. The info was added in the text
Q6) Add figure 1 as a study flow chart for the readers. (Major Compulsory Revision).
A6) The flow chart was added
Q7) The Discussion section is a rehashing of the results. It does not appear that the authors include much interpretation of what the study findings mean for clinical practice or research. (Major Compulsory Revision)
The authors beleve that the discussion s the summery of the results generalization, for example the the syndesmosis micro-mobility section is a generalitation of faster recovery of patients treated with TightRope and was explained by the early recovery in gait analisys
Finally, the conclusión is weak and too long. (Major Compulsory Revision)
A7) Conclusion has been modified
Round 2
Reviewer 1 Report
The reviewer thanks you for submitting the revised version. Some noted questions could be answered in the meantime and hints were partially implemented.
However, please allow my critical comments again:
I recommend the authors not to keep the title so general. Here the impression is given that a metha-analysis of the two procedures is carried out, whereas it is actually about a specific scientific question.
Overall, the publication now reads somewhat more fluently, although basic rules of scientific writing are not yet fully implemented.
For example, company names are mentioned in the institution. Here I recommend to choose a paraphrase, e.g. "suture button device", in order not to give the impression that it would be a study financed by the industry. It is also debatable whether the name of the hospital should be mentioned.
In the discussion section the own results are still not discussed but only an overview of the scientific literature is given. In the current form it is not a discussion but a literature review.
In detail, the statement why isolated syndesmosis injuries were not considered is missing. They would be particularly suitable for comparing the two stabilization methods. Furthermore, the isolated injuries are not mentioned in the exclusion criteria.
The question about the ethics vote remained unanswered as well as the question how many (number of) surgeons/postoperative care providers were involved.
Since X-ray controls were performed at all measurement points and apparently the patients were also seen clinically, the question of the study design must also be asked. Why was a prospective study approach not chosen?
The fact that in this large TightRope group there was apparently no nodal irritation or secondary extension of the TightRope or radiologically visible osteolysis is not consistent with the existing literature and our own experience. It was precisely because of the knot irritations that the company promptly introduced the "knotless TighRope" to the market, as some studies were able to demonstrate this problem.
Screw fractures also do not seem to have been present or are not mentioned. In this context, the postoperative follow-up treatment still remains unclear: partial loading, full loading for how long? No statement is made on this.
Why the gait analysis was only performed twice in the 2 years also remains unclear.
In summary, I unfortunately still see many methodological weaknesses and ambiguities in this publication. In addition, I miss the critical scientific discussion with the own results.
Author Response
The review thanks you for submitting the revised version. Some noted questions could be answered in the meantime and hints were partially implemented.
However, please allow my critical comments again:
Q1) I recommend the authors not to keep the title so general. Here the impression is given that a metha-analysis of the two procedures is carried out, whereas it is actually about a specific scientific question.
A1) As you suggested the title was partially re-written
Q2) Overall, the publication now reads somewhat more fluently, although basic rules of scientific writing are not yet fully implemented.
For example, company names are mentioned in the institution. Here I recommend to choose a paraphrase, e.g. "suture button device", in order not to give the impression that it would be a study financed by the industry. It is also debatable whether the name of the hospital should be mentioned.
A2) The requested modifies were perfomed.
Q3) In the discussion section the own results are still not discussed but only an overview of the scientific literature is given. In the current form it is not a discussion but a literature review.
A3) Further paragraphs were added in the section
Q4) In detail, the statement why isolated syndesmosis injuries were not considered is missing. They would be particularly suitable for comparing the two stabilization methods. Furthermore, the isolated injuries are not mentioned in the exclusion criteria.
A4) Aim of the study was to compare the fixation systems in ankle fractures, for this reason the isolated syndesmosis were excluded.
Q5) The question about the ethics vote remained unanswered as well as the question how many (number of) surgeons/postoperative care providers were involved.
Since X-ray controls were performed at all measurement points and apparently the patients were also seen clinically, the question of the study design must also be asked. Why was a prospective study approach not chosen?
A5) the study was a retrospective case series, the authors are developing a prospective trials and futher evidences were provide. For retrospective studies our istitution does require ethics commite approval3. Four different surgeons the same foot and ankle team.
Q6) The fact that in this large TightRope group there was apparently no nodal irritation or secondary extension of the TightRope or radiologically visible osteolysis is not consistent with the existing literature and our own experience. It was precisely because of the knot irritations that the company promptly introduced the "knotless TighRope" to the market, as some studies were able to demonstrate this problem.
A6) We agree with your comments, in our series there is no nodal irritation or extension we added your concept in discussion
Q7) Screw fractures also do not seem to have been present or are not mentioned. In this context, the postoperative follow-up treatment still remains unclear: partial loading, full loading for how long? No statement is made on this.
A7) Requested info was added in Methods
Q8) Why the gait analysis was only performed twice in the 2 years also remains unclear.
A8) The gait analysis data were recorded by our istitution physical theraphy unit. Their protocol do not provide any other data.
In summary, I unfortunately still see many methodological weaknesses and ambiguities in this publication. In addition, I miss the critical scientific discussion with the own results.
Reviewer 2 Report
I am happy with the paper as it stands. Congratulations.
Author Response
Thank you for your comments